# Daily Coffee and Green Tea Consumption Is Inversely Associated with Body Mass Index, Body Fat Percentage, and Cardio-Ankle Vascular Index in Middle-Aged Japanese Women: A Cross-Sectional Study

**DOI:** 10.3390/nu12051370

**Published:** 2020-05-11

**Authors:** Yuka Yonekura, Masakazu Terauchi, Asuka Hirose, Tamami Odai, Kiyoko Kato, Naoyuki Miyasaka

**Affiliations:** 1Department of Obstetrics and Gynecology, Tokyo Medical and Dental University, Yushima 1-5-45, Bunkyo, Tokyo 113-8510, Japan; 160992ms@tmd.ac.jp (Y.Y.); a-kacrm@tmd.ac.jp (A.H.); n.miyasaka.gyne@tmd.ac.jp (N.M.); 2Department of Women’s Health, Tokyo Medical and Dental University, Yushima 1-5-45, Bunkyo, Tokyo 113-8510, Japan; odycrm@tmd.ac.jp (T.O.); kiyo.crm@tmd.ac.jp (K.K.)

**Keywords:** menopause, atheromatous arteriosclerosis, metabolic syndrome, obesity, caffeine, polyphenol

## Abstract

This study aimed to investigate the links between coffee (CF)/green tea (GT) consumption and body composition/cardiovascular parameters in middle-aged Japanese women. We conducted a cross-sectional study of 232 Japanese women aged 40–65 years who had been referred to the menopause clinic of Tokyo Medical and Dental University Hospital between November 2007 and August 2017. Body composition, cardiovascular parameters, and CF/GT consumption frequency were evaluated on their initial visits, using a body composition analyzer, vascular screening system, and brief-type self-administered diet history questionnaire, respectively. We investigated the associations between variables using multivariate logistic regression. After adjustment for age, menopausal status, and other factors, daily CF consumption was inversely associated with high body mass index (BMI) (adjusted odds ratio, 0.14; 95% confidence interval, 0.14–0.96) and body fat percentage (BF%) (0.33; 0.14–0.82), and daily GT consumption with high BF% (0.36; 0.14–0.96). Daily CF + GT consumption was also inversely associated with high BMI (0.15; 0.05–0.50) and BF% (0.30; 0.12–0.74). In pre- and perimenopausal women, daily CF + GT consumption was inversely associated with high cardio-ankle vascular index (CAVI) (0.05; 0.003–0.743). In conclusion, daily CF/GT consumption was inversely associated with high BMI, BF%, and CAVI in middle-aged Japanese women.

## 1. Introduction

In the menopausal transition (perimenopausal) and postmenopausal periods, a reduction in estrogen levels not only causes various symptoms such as hot flashes, night sweats, vaginal dryness, depression, anxiety, and insomnia, but also changes fat distribution from subcutaneous to visceral adiposity, which is one of the major risk factors of atheromatous arteriosclerosis. Atherosclerosis is a chronic inflammatory disease [1]. It is the main pathophysiological cause of cardiovascular diseases (CVDs), which are the biggest causes of death worldwide. According to the World Health Organization (WHO), 17.9 million people died of CVDs in 2016 [2]. Considering that the prevalence of CVDs steeply increases after menopause [3,4,5,6], it is important to lower the risk of atheromatous arteriosclerosis in middle-aged women.

Healthy lifestyles, including a well-balanced diet, physical activity, quitting smoking, and alcohol moderation, are well-known preventive measures of atherosclerosis. It has also been reported that beverages such as coffee (CF) and green tea (GT) are effective in lowering cardiovascular risks. A couple of human trials have revealed the anti-obesity effects of CF and GT extracts [7,8]. CF and GT consumption is also associated with a lower risk of stroke [9]. Furthermore, a pilot study reported that CF extract improves arterial stiffness in healthy Japanese men [10], and a meta-analysis showed that GT intake improves cardiovascular risk markers, including systolic blood pressure [11]. However, the effects of CF and GT intakes in middle-aged women, who are threatened by increasing cardiovascular risks, have yet to be elucidated. Cardio-ankle vascular index (CAVI) is one of the noninvasive measures used as an index of atherosclerosis. CAVI is superior to other measures in that we could also measure other cardiovascular parameters at the same time, and it could well represent the arterial stiffness from the origin of aorta to ankle [12,13,14]. The present study aimed to investigate the associations between CF/GT consumption and body composition/cardiovascular parameters, including CAVI, in middle-aged Japanese women.

## 2. Materials and Methods

### 2.1. Study Population

In this cross-sectional study, we analyzed the first-visit records of 232 Japanese women who were enrolled in the Systematic Health and Nutrition Education Program at the menopause clinic of Tokyo Medical and Dental University Hospital, Japan, from November 2007 to August 2017. All women who enrolled in this program had attended our clinic to treat menopausal symptoms. The collected data included age, menopausal status, body composition parameters, cardiovascular parameters, lifestyle factors, and detailed dietary habits. These data were collected by physicians and nutritionists on their initial visits. The inclusion criteria were those aged between 40 and 65 years, reported their menopausal status, and completed the brief-type self-administered diet history questionnaire (BDHQ). The exclusion criteria were those who had been treated with menopausal hormone therapy, antihyperlipidemic drugs, and antidiabetic agents, whose body mass index (BMI) was <16 kg/m^2^ or >35 kg/m^2^, and who answered in the BDHQ that they drank black tea ≥ 1 cup/day. 

The research protocol was reviewed and approved by the Tokyo Medical and Dental University Review Board (approval number: 774), and written informed consent was obtained from all participants. The study was conducted in accordance with the Declaration of Helsinki.

### 2.2. Measurements

#### 2.2.1. Menopausal Status

Participants were classified as premenopausal if they had regular menstrual cycles; perimenopausal if they had a menstrual period within the past 12 months but had missed periods or had an irregular cycle in the past 3 months; and postmenopausal if they had no menstruation in the past 12 months.

#### 2.2.2. Physical Assessments and Cardiovascular Parameters

Body composition parameters, including height, weight, BMI, fat mass, muscle mass, and lean body mass, were measured using a body composition analyzer (MC190-EM; Tanita, Tokyo, Japan). Resting energy expenditure was calculated based on respiratory volume using a portable indirect calorimeter (Metavine-N VMB-005 N; Vine, Tokyo, Japan). Additionally, cardiovascular parameters, including systolic and diastolic blood pressure, heart rate, cardio-ankle vascular index (CAVI), and ankle-brachial pressure index (ABI), were assessed using a vascular screening system (VS-1000; Fukuda Denshi, Tokyo, Japan). Plasma blood sugar level was also measured. Blood examination was conducted in accordance with the guidelines on internal and external quality control defined by the Japanese Ministry of Health, Labor, and Welfare.

#### 2.2.3. Lifestyle Characteristics

Participants underwent a medical interview for lifestyle factors, which included the frequency of alcohol consumption (never, sometimes, every day), smoking (no, yes), and regular exercise habits (none, <3 times/week, ≥3 times/week).

#### 2.2.4. Dietary Habits

Dietary habits were assessed using the BDHQ. The BDHQ asked about the consumption frequencies of selected food and beverage items commonly consumed in Japan. Based on the participants’ responses to the questionnaire, an ad-hoc computer algorithm estimated the amounts of 96 nutrients consumed during the previous month. The choices for the frequencies of CF and GT consumption were as follows: 0, <1, 1, 2–3, or 4–6 times/week, or 1, 2–3, or ≥4 cups/day. Energy intake was calculated based on the responses to the BDHQ.

### 2.3. Factors Associated with Body Composition/Cardiovascular Parameters

We dichotomized CF and GT consumption as low (<1 cup/day) and high (≥1 cup/day), and the women were divided into four groups according to their consumption frequency; (i) control group, who consumed < 1 cup/day of both CF and GT; (ii) CF group, who consumed ≥ 1 cup/day of CF alone; (iii) GT group, who consumed ≥ 1 cup/day of GT alone; and (iv) CF + GT group, who consumed ≥ 1 cup/day of both CF and GT. To determine the background factors which could be confounding, women in these four groups were compared for background factors, including age, menopausal status, height, basal metabolism, energy intake, and lifestyle characteristics. The factors that significantly differed (*p* < 0.05) among the groups were selected as explanatory variables for a multivariate logistic regression analysis.

### 2.4. Statistical Analysis

The required total sample size was estimated at 233, as calculated from the number of predictive variables, events per variable, and event incidence rate of 7, 10, and 0.30, respectively. First, to determine which parameters were associated with CF and GT consumption, we compared body composition and cardiovascular parameters among four groups. Next, we conducted a multivariate logistic regression analysis to find out whether CF/GT consumption was independently associated with the selected body composition/cardiovascular parameters. We examined the association, adjusting for the extracted background factors. When CF/GT consumption frequency retained a significant association (*p* < 0.05) with the selected body composition/cardiovascular parameters in the final multivariate model, we considered that CF/GT consumption was associated with the selected body composition/cardiovascular parameters in Japanese middle-aged women. Statistical analysis, including a one-way analysis of variance (ANOVA) with Tukey’s multiple comparison test and a chi-squared test, was performed with GraphPad Prism version 5.02 (GraphPad Software, San Diego, CA, USA). The multivariate logistic regression analysis was performed with JMP version 14 (SAS Institute Inc., Cary, NC, USA). A *p*-value of <0.05 was considered statistically significant.

## 3. Results

The average age of the participants was 51.6 ± 5.0 years (mean ± standard deviation). The number of women who were classified into the control, CF, GT, and CF + GT groups was 39 (16.8%), 47 (20.3%), 76 (32.8%), and 70 (30.2%), respectively.

We first compared body composition, cardiovascular parameters, and background characteristics among the four groups using a one-way ANOVA with Tukey’s multiple comparison test and a chi-squared test. The body composition and cardiovascular parameters that statistically differed (*p* < 0.05) at the univariate level among the four groups were height (cm), body weight (kg), BMI (kg/m^2^), body fat (%), and fat mass (kg) (Table 1).

Of these parameters, body weight, BMI, body fat percentage (BF%), and fat mass showed a similar decreasing tendency in the order of control > CF, GT > CF + GT. Considering that body weight and fat mass were dependent on height, we decided to limit further analysis of the associations between BMI/BF% and daily CF/GT consumption. Regarding background characteristics related to BMI and BF%, we selected age, menopausal status, energy intake, smoking habit, frequency of alcohol consumption, and regular exercise habit, which statistically differed among the four groups (*p* < 0.05, Table 1). We performed multivariate logistic regression analysis to determine the independent relationships between daily CF/GT consumption and high BMI (≥25 kg/m^2^)/high BF% (≥30%).

After adjustment for age and menopausal status (Model 2), and also for selected background characteristics (Model 3), both CF consumption and CF + GT consumption of > 1 cup/day exhibited a significant inverse relationship with high BMI (CF: adjusted odds ratio (OR], 0.14; 95% confidence interval (CI], 0.05–0.46; *p* < 0.01; and CF + GT: OR, 0.15; 95% CI, 0.05–0.50; *p* < 0.01), while the consumption of all three, CF, GT, and CF + GT, was inversely associated with high BF% (CF: OR, 0.33; 95% CI, 0.14–0.82; *p* < 0.05; GT: OR, 0.36; 95% CI, 0.14–0.96; *p* < 0.05; and CF + GT: OR, 0.30; 95% CI, 0.12–0.74; *p* < 0.01) (Table 2).

We also conducted subgroup analyses of the associations between CF/GT consumption and body composition/cardiovascular parameters in pre-/perimenopausal and postmenopausal women in the same manner. At the univariate level, a significant difference was not observed with regard to body composition. As for cardiovascular parameters, CAVI significantly differed among the four groups in pre-/perimenopausal women and exhibited a decreasing trend. Using CAVI = 8.0 as the cutoff value, we performed multivariate logistic regression analysis to investigate the relationship between daily CF/GT consumption and high CAVI (≥8.0). After adjustment for age (Model 2), and also for selected background characteristics (Model 3), daily CF + GT consumption was inversely associated with high CAVI (OR, 0.05; 95% CI, 0.003–0.743; *p* < 0.05) (Table 3).

## 4. Discussion

In this cross-sectional analysis of 232 middle-aged Japanese women who attended our menopause clinic, the daily intake of CF/GT was shown to be inversely associated with high BMI and BF%. Moreover, in pre- and perimenopausal women, the daily intake of both CF and GT was inversely associated with high CAVI.

CF comprises many components with pharmacologic effects, such as caffeine and chlorogenic acid (CGA). CGA is one of the polyphenols that are abundant in green CF beans and is considered to be the most active compound [15,16,17]. Some studies showed that CGA has the effect of suppressing the accumulation of body fat. A randomized, double-blind trial showed that the daily consumption of CF rich in CGA for 12 weeks by overweight Japanese adults lowered the visceral fat area, total abdominal fat area, BMI, and waist circumference [7]. It was also reported that coffee reduced the accumulation of lipids during adipocytic differentiation of 3T3-L1 preadipocytes and inhibited the expression of peroxisome proliferator-activated receptor γ, which controls the differentiation of adipocytes [18]. Another underlying mechanism was suggested in human trials. These trials reported that CGA consumption increased postprandial energy expenditure and fat utilization in healthy humans [19,20]. It is plausible that CF ingredients may promote fat oxidation and suppress fat differentiation in humans.

In addition, CGA also has anti-atherosclerotic properties [21]. A single-blind, randomized, placebo-controlled, crossover trial showed that a single intake of CF with a high content of CGA and a low content of hydroxyhydroquinone, which oxidizes CGA and reduces its function, improved postprandial endothelial dysfunction by decreasing oxidative stress [22]. A recent meta-analysis of randomized clinical trials reported the antihypertensive effect of CGA [23]. Moreover, a randomized controlled trial showed that coffee rich in caffeoylquinic acid, one of the CGAs, improved the levels of plasma total cholesterol, triglycerides, and low-density lipoprotein cholesterol [24]. These results indicate that CGA could be effective in preventing atherosclerosis.

GT also contains several green tea catechins (GTCs), such as epicatechin, epicatechin gallate, epigallocatechin, and epigallocatechin gallate (EGCG), as well as their thermal isomers, such as catechin, catechin gallate, gallocatechin, and gallocatechin gallate [25]. Of these catechins, EGCG is the most abundant in GT infusions and is considered to be the most active compound [25,26]. A pooled analysis of six human trials demonstrated that the consumption of beverages containing GTCs, mainly EGCG, for 12 weeks significantly reduced the total fat area, visceral fat area, BMI, and waist circumference [27]. A cell-based experiment partly revealed the mechanism with which EGCG inhibits 3T3-L1 preadipocyte differentiation, activates AMPK, and decreases fat accumulation [28]. GTCs could also have protective effects on endothelial cells as other antioxidants do, by inhibiting the adhesion of monocytes [29,30]. In another cell-based study, EGCG was shown to suppress the mRNA expression of monocyte chemotactic protein-1, which accelerates the progress of atherosclerosis by promoting the adhesion of monocytes [31]. These findings indicate that GT may be a potential therapeutic agent for the prevention of atherosclerosis.

In this study, the combined intake of CF and GT was inversely associated with high CAVI in pre- and perimenopausal women; however, the intake of CF or GT alone did not show any inverse relationship. These results suggest the additive effect of CF and GT in lowering CAVI. A large cohort study in Japan showed that the consumption of ≥4 cups/day of GT and a combined intake of ≥1 cup/day of CF and ≥2 cups/day of GT contributed to a risk reduction in CVD and stroke [11]. Considering that a cup of GT contains about 112 mg EGCG and a cup of CF contains about 160 mg CGA [32], it is possible that a daily intake of 400 mg of polyphenols is effective in preventing CVD. Although the mechanism underlying the combined effect remains unclear, the different antioxidants in CF and GT may additively strengthen the beneficial effects on body composition and cardiovascular parameters. In another population-based prospective study in Japan, the researchers found that dietary polyphenol intake is inversely associated with CVD [33]. To corroborate the association between polyphenols and CVD risk, more comprehensive studies are warranted.

In this study, the daily intake of CF/GT was not inversely associated with high CAVI in postmenopausal women. After menopause, a reduction in estrogen levels causes rapid endothelial dysfunction, which could not be overcome by the moderate effects of CF and GT on vascular endothelial function.

This study has several limitations. First, the study population was relatively small and consisted only of middle-aged Japanese women who attended our menopause clinic. Therefore, generalizing our results to a wider population is difficult. Second, we did not consider polyphenols from foods other than CF and GT in this study. Finally, the cross-sectional design of this study prevented the determination of a causal relationship between the daily intake of CF/GT and lower BMI, BF%, and CAVI.

Nevertheless, this study has several strengths. Background factors associated with body composition and cardiovascular parameters were well investigated, including energy intake and expenditure, and lifestyles. To corroborate the observed association between the daily intake of CF/GT and body composition, we recommend prospective studies that enroll a higher number of obese and hypertensive women and evaluate both daily intake and serum levels of the polyphenols.

## 5. Conclusions

This study revealed that daily CF and GT consumption was inversely associated with high BMI and BF% in middle-aged Japanese women. In pre- and perimenopausal women, the daily consumption of both CF and GT was inversely associated with high CAVI. These results suggest that the consumption of CF and GT could help women to keep fit and prevent atherosclerosis.

## Figures and Tables

**Table 1 nutrients-12-01370-t001:** Comparison of background characteristics and daily consumption of coffee (CF) and green tea (GT).

	Control (n = 39)	CF (n = 76)	GT (n = 47)	CF + GT (n = 70)	*p*-Value
Age and menopausal status					
Age (years), mean (SD)	51.2 (6.1)	50.4 (3.8)	52.0 (5.2)	52.8 (5.3)	0.027 ^a^
Menopausal status					
Premenopause	11 (28.2)	31 (40.8)	7 (14.9)	17 (24.3)	
Perimenopause	8 (20.5)	20 (26.3)	6 (12.8)	11 (15.7)	
Postmenopause	20 (51.3)	25 (32.9)	34 (72.3)	42 (60.0)	0.002 ^b^
Body composition, mean (SD)					
Height (cm)	157.8 (6.9)	158.7 (5.1)	155.5 (6.4)	157.6 (4.9)	0.028 ^a^
Body weight (kg)	57.4 (12.7)	53.9 (8.1)	52.4 (9.6)	52.1 (7.1)	0.022 ^a^
Body mass index (kg/m^2^)	23.0 (4.4)	21.4 (3.2)	21.6 (3.5)	21.0 (2.7)	0.027 ^a^
Body fat (%)	30.5 (8.5)	26.8 (7.0)	27.2 (7.5)	25.8 (6.6)	0.014 ^a^
Fat mass (kg)	18.5 (8.9)	14.9 (6.2)	14.8 (6.7)	13.8 (5.1)	0.005 ^a^
Muscle mass (kg)	36.7 (4.2)	36.7 (2.8)	35.4 (3.7)	36.1 (2.9)	0.140 ^a^
Lean body mass (kg)	38.9 (4.5)	39.0 (3.1)	37.6 (4.0)	38.3 (3.2)	0.140 ^a^
Cardiovascular parameters, mean (SD)					
Systolic blood pressure (mmHg)	131 (21)	126 (18)	126 (20)	125 (16)	0.371 ^a^
Diastolic blood pressure (mmHg)	83 (14)	80 (11)	81 (13)	80 (11)	0.575 ^a^
Pulse rate (beats/minute)	66 (16)	64 (11)	64 (11)	64 (11)	0.822 ^a^
Blood sugar level (mg/dL)	98 (23)	96 (8)	100 (14)	95 (8.4)	0.319 ^a^
Cardio-ankle vascular index	7.78 (0.84)	7.45 (0.70)	7.54 (0.65)	7.50 (0.71)	0.178 ^a^
Ankle-brachial pressure index	1.11 (0.06)	1.11 (0.06)	1.12 (0.07)	1.10 (0.05)	0.475 ^a^
Basal metabolism, mean (SD)					
Resting energy expenditure (kcal/day)	1669 (527)	1628 (385)	1618 (419)	1512 (393)	0.220 ^a^
Dietary intake, mean (SD)					
Energy intake (kcal/day)	1490 (532)	1638 (394)	1735 (544)	1786 (468)	0.013 ^a^
Lifestyle characteristics, number (%)					
Smoking habit					
Yes	2 (5.1)	10 (13.2)	0 (0)	4 (5.7)	
No	37 (94.9)	66 (86.8)	47 (100)	66 (94.3)	0.037 ^b^
Frequency of alcohol consumption					
Every day	3 (7.7)	13 (17.1)	7 (14.9)	2 (2.9)	
Sometimes	14 (35.9)	28 (36.8)	8 (17.0)	21 (30.0)	
Nondrinker	22 (56.4)	35 (46.1)	32 (68.1)	47 (67.1)	0.015 ^b^
Regular exercise habit					
≥3 times/week	1 (2.6)	8 (10.5)	6 (12.8)	5 (7.1)	
1–2 times/week	18 (46.2)	13 (17.1)	12 (25.5)	18 (25.7)	
None	20 (51.3)	55 (72.4)	29 (61.7)	47 (67.1)	0.039 ^b^

^a^ One-way analysis of variance. ^b^ Chi-squared test. CF, coffee; GT, green tea; SD, standard deviation.

**Table 2 nutrients-12-01370-t002:** Multivariate analysis of the associations of daily coffee (CF) and green tea (GT) consumption with high body mass index (BMI) and body fat percentage (BF%).

Model	Group	BMI ≥ 25			BF% ≥ 30		
		OR	95% CI	*p*-Value	OR	95% CI	*p*-Value
Model 1	CF	0.26	0.10–0.72	<0.01	0.42	0.19–0.93	<0.05
	GT	0.53	0.20–1.44	0.2	0.49	0.20–1.15	0.1
	CF + GT	0.21	0.07–0.62	<0.01	0.34	0.15–0.78	<0.05
Model 2	CF	0.25	0.09–0.69	<0.01	0.43	0.19–0.95	<0.05
	GT	0.57	0.21–1.56	0.3	0.46	0.19–1.12	0.09
	CF + GT	0.21	0.07–0.64	<0.01	0.35	0.15–0.80	<0.05
Model 3	CF	0.14	0.05–0.46	<0.01	0.33	0.14–0.82	<0.05
	GT	0.38	0.12–1.18	0.09	0.36	0.14–0.96	<0.05
	CF + GT	0.15	0.05–0.50	<0.01	0.30	0.12–0.74	<0.01

BMI, body mass index; BF%, body fat percentage; CI, confidence interval; OR, odds ratio CF, coffee; GT, green tea.

**Table 3 nutrients-12-01370-t003:** Multivariate analysis of the associations of daily coffee (CF) and green tea (GT) consumption with high cardio-ankle vascular index (CAVI) in pre-/perimenopausal women.

Model	Group	CAVI ≥ 8.0		
		OR	95% CI	*p*-Value
Model 1	CF	0.47	0.12–1.94	0.3
	GT	0.23	0.02–2.39	0.2
	CF + GT	0.21	0.03–1.33	0.1
Model 2	CF	0.38	0.08–1.91	0.2
	GT	0.13	0.01–1.66	0.1
	CF + GT	0.12	0.02–1.00	<0.05
Model 3	CF	0.37	0.05–2.53	0.3
	GT	0.11	0.01–2.27	0.2
	CF + GT	0.05	0.003–0.743	<0.05

CAVI, cardio-ankle vascular index; CI, confidence interval; OR, odds ratio; CF, coffee; GT, green tea.

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
