# Peer review of "Daily Coffee and Green Tea Consumption Is Inversely Associated with Body Mass Index, Body Fat Percentage, and Cardio-Ankle Vascular Index in Middle-Aged Japanese Women: A Cross-Sectional Study"

_nutrients, 2020, doi:10.3390/nu12051370_

Round 1
Reviewer 1 Report
This cross-sectional study investigated middle-aged Japanese women (232), grouped control, daily coffee, and green tea alone or combined consumption is inversely associated with body mass index, body fat percentage, and cardio-ankle vascular. The authors concluded that daily CF/GT consumption was inversely associated with high BMI, BF%, and CAVI in middle-aged Japanese women. Thus, CF and GT consumption may help middle-aged women to keep fit and prevent atherosclerosis.
This cross-sectional observational study well designed, overall, this manuscript written well, but few queries need to be addressed.
Comments:
- Atherosclerosis well documented with insulin resistance and diabetes. Why authors preferred BMI, BF percentage and CAVI; instead of HOMA-IR (insulin resistance index). Hence, details of glycemia must be incorporated to validate this point.
- The importance of the CAVI parameter must be included in the introduction compared with other non-invasive techniques or parameters. It is suggestive of adding the recent review (PMID:31618136) to show the other essential methods used to measure cardiac dysfunction given obesity.
- It is advisable to provide a separate table to show pre, peri, post-menopause stage differences in BMI; since middle age menopause condition and post-menopause increases visceral fat.
- Did the authors measure estradiol and testosterone in those subjects? If so, include those, it would be interesting to see how CF, GT affect the circulating hormones.
- Discussion portions talk about the phytoactive compounds present in CF/GT and its antioxidant properties; maybe few recent publications may include (PMID:30160165, PMID:30582899, PMID:29626298) to support the how antioxidant system help in preventing endothelial dysfunctions /atherosclerosis in obese or diabetic conditions.
- Check abbreviations, typos, and minor grammar syntax errors throughout the manuscript.
Reviewer 2 Report
The present manuscript is about „Daily coffee and green tea consumption is inversely associated with body mass index, body fat percentage, and cardio-ankle vascular index in middle-aged Japanese women: a cross-sectional study“. The
manuscript has some major flaws. The study design should be described in more detail. Also in the abstract the place, time, duration, design of the study are mentioned. The study participants should be described better. It is not clear from the study when the parameters of the study of the study population were collected. Was there a baseline? How do you get the connection that the body weight after drinking coffee and green tea improved the cardiocascular parameters of the study population? Was the body weight of the study participants measured regularly? Have the other parameters been measured regularly? The conduct of the study should describe how the parameters were collected. The mention of animal testing should be deleted from the entire text. The abbreviations in the tables should be explained below the table. The confounding factors of the study should be mentioned in the method. The secondary diseases of the study participants could have led to a distortion of the parameters. The secondary diseases of the study population were not included in the study.
Round 2
Reviewer 2 Report
The revised work is available. However, the suggestions for improvement were not considered.
